# Time to Load Up–Resistance Training Can Improve the Health of Women with Polycystic Ovary Syndrome (PCOS): A Scoping Review

**DOI:** 10.3390/medsci10040053

**Published:** 2022-09-22

**Authors:** Chris Kite, Elizabeth Parkes, Suzan R. Taylor, Robert W. Davies, Lukasz Lagojda, James E. Brown, David R. Broom, Ioannis Kyrou, Harpal S. Randeva

**Affiliations:** 1School of Public Health Studies, Faculty of Education, Health and Wellbeing, University of Wolverhampton, Wolverhampton WV1 1LY, UK; 2Warwickshire Institute for the Study of Diabetes, Endocrinology and Metabolism (WISDEM), University Hospitals Coventry and Warwickshire NHS Trust, Coventry CV2 2DX, UK; 3Centre for Sport, Exercise and Life Sciences, Research Institute for Health & Wellbeing, Coventry University, Coventry CV1 5FB, UK; 4Chester Medical School, University of Chester, Shrewsbury SY3 8HQ, UK; 5Clinical Evidence Based Information Service (CEBIS), University Hospitals Coventry and Warwickshire NHS Trust, Coventry CV2 2DX, UK; 6School of Biosciences, College of Health and Life sciences, Aston University, Birmingham B4 7ET, UK; 7Aston Medical School, College of Health and Life Sciences, Aston University, Birmingham B4 7ET, UK; 8Warwick Medical School, University of Warwick, Coventry CV4 7AL, UK; 9Laboratory of Dietetics and Quality of Life, Department of Food Science and Human Nutrition, School of Food and Nutritional Sciences, Agricultural University of Athens, 11855 Athens, Greece

**Keywords:** strength training, lifestyle, metabolism, hormones, quality of life, women’s health

## Abstract

Background: Guidelines for the management of polycystic ovary syndrome (PCOS) focus on lifestyle changes, incorporating exercise. Whilst evidence suggests that aerobic exercise may be beneficial, less is known about the effectiveness of resistance training (RT), which may be more feasible for those that have low fitness levels and/or are unable to tolerate/participate in aerobic exercise. Objectives: To identify the available evidence on RT in women with PCOS and to summarise findings in the context of a scoping review. Eligibility criteria: Studies utilising pre-post designs to assess the effectiveness of RT in PCOS; all outcomes were included. Sources of evidence: Four databases (PubMed, CENTRAL, CINAHL and SportDiscus) were searched and supplemented by hand searching of relevant papers/reference lists. Charting methods: Extracted data were presented in tables and qualitatively synthesised. Results: Searches returned 42 papers; of those, 12 papers were included, relating to six studies/trials. Statistical changes were reported for multiple pertinent outcomes relating to metabolic (i.e., glycaemia and fat-free mass) and hormonal (i.e., testosterone and sex hormone-binding globulin) profiles. Conclusions: There is a striking lack of studies in this field and, despite the reported statistical significance for many outcomes, the documented magnitude of changes are small and the quality of the evidence questionable. This highlights an unmet need for rigorously designed/reported and sufficiently powered trials.

## 1. Introduction

Polycystic ovary syndrome (PCOS) is the most common endocrine disorder in reproductive-aged women which, depending on the applied diagnostic criteria, affects 15–20% of this female population [1,2]. Following exclusion of other relevant conditions, PCOS is typically diagnosed based on the presence of at least two out of three criteria; that is, chronic anovulation, hyperandrogenism (clinical and/or biochemical) and polycystic ovaries as identified by ultrasound [3]. In addition, women with PCOS are also at an increased risk of cardiometabolic complications, particularly obesity, insulin resistance, type 2 diabetes, hypertension, non-alcoholic fatty liver disease, and obstructive sleep apnoea [4,5,6,7,8,9]. Furthermore, women with PCOS are also at a high risk of psychological comorbidity, exhibiting higher prevalence of coexisting anxiety and/or depression [10,11,12,13,14], and impairments to their overall quality of life (QoL) compared to women without PCOS [15,16]. As there is currently no curative treatment for this syndrome, management of women with PCOS aims to alleviate the clinical manifestations, whilst lowering the related risk of cardiometabolic morbidity [17]. Accordingly, first line management recommendations focus on lifestyle changes aiming to increase physical activity (PA) and improve dietary habits [18]. 

Lifestyle interventions which incorporate exercise have been shown to improve many of these cardiometabolic health-related outcomes in a range of general and patient populations [19,20,21]. Given this existing evidence, lifestyle interventions represent an effective strategy to support improvements in the health and wellbeing of women with PCOS. Indeed, our previous systematic review and meta-analysis of randomised controlled trials (RCTs) that utilised exercise interventions in women with PCOS [22] revealed statistically favourable effects for several health-related outcomes. Based on these data, women with PCOS who completed an exercise intervention had improved fasting insulin, insulin resistance, total cholesterol, low-density lipoprotein cholesterol (LDL-C), triglycerides, cardiorespiratory fitness, waist circumference and body fat percentage, when compared to those receiving no intervention. Of note, the most successful exercise interventions were those which were supervised, of a shorter duration, and which included only aerobic exercise [22]. However, the certainty of the existing evidence is either low or very low, and warrants a cautious approach when interpreting these findings since the included published studies had small sample sizes and a high risk of bias, whilst many outcomes had modest effects and wide 95% confidence intervals (CI) [23].

Regarding exercise modality, it is notable that only three of the included RCTs in our systematic review [24,25,26] had a resistance training arm as part of their intervention, with data from just 25 participants in total. This striking paucity of RCTs including women with PCOS completing resistance training interventions is surprising, particularly given the benefits reported in other populations. Resistance training is a health behaviour which is critical to health and is included in many international PA guidelines [27,28,29,30]. An overview of systematic reviews, including data from 382,627 participants, reported that when adults engage in resistance training, risk of mortality and myocardial infarction are lowered, whilst blood pressure, muscular strength and physical functioning are also improved [31]. Moreover, a systematic review by Gordon and colleagues [32] reported improved glycaemic control and insulin sensitivity following resistance training interventions in patients with type 2 diabetes. Further systematic reviews have reported a significant positive relationship between participation in resistance training and QoL [33], fatigue levels [33], improvement of depressive symptoms [34], sleep quality [35], body composition [36] and hepatic steatosis [37]. Despite the small number of women with PCOS represented in our previous meta-analysis [22], we also reported statistical improvements to fasting insulin, high-density lipoprotein cholesterol (HDL-C) and waist circumference when those specifically completing a resistance training intervention were compared to control.

Overall, there is an increasing number of studies investigating the effects of exercise in PCOS, but there appears to be a lack of well-designed/reported studies which focus on the effects of resistance training. Given that women with PCOS have markedly lower levels of cardiorespiratory fitness compared to healthy controls [38], resistance training may represent a more feasible alternative to aerobic exercise for those with low cardiorespiratory fitness levels, or for those who are unable to tolerate/participate in aerobic exercise. Accordingly, it is important to summarise this body of evidence in order to better understand the effects of resistance training on the health of women with PCOS, and to better inform the relevant clinical practice. Therefore, the aims of this scoping review are to examine the extent and nature of the available evidence on resistance training in women with PCOS and, from this evidence, to summarise key findings for all included outcomes.

## 2. Materials and Methods

This scoping review is reported based on the guidelines of the Preferred Reporting Items for Systematic Reviews and Meta-Analyses extension for Scoping Reviews (PRISMA-ScR) checklist (Appendix A) [39]. The protocol for this review was adapted from a Master of Science dissertation project (EP) conducted at the University of Chester.

### Search Methods for Identification of Studies

Table 1 presents the eligibility criteria for studies to be included in this scoping review. Only studies that include at least one study arm in which reproductive-aged women with PCOS complete a resistance/strength training intervention were eligible for inclusion. Whilst women with PCOS are the focus of this study, studies that have used a case–control design (e.g., comparing women with PCOS to healthy women without PCOS) were eligible for inclusion. Eligible studies had to employ an intervention design, and report pre- and post-intervention data that measure the chronic effects of resistance training exercise in women with PCOS; accordingly, a variety of study designs were eligible for inclusion. We defined resistance/strength training as a method of conditioning in which an individual works against a resistive load to enhance health, fitness and/or performance [40]. 

The databases searched were PubMed, CENTRAL (in the Cochrane Library), CINAHL and SportDiscus (via EBSCOhost). A search algorithm was developed for PubMed (Table 2), which was adapted for the additional databases. Searches were completed in July 2022 with no time limit specified for study inclusion (no limit was applied for the range of publication years). There were no language restrictions placed on the search, but only fully published studies, that had undertaken peer review were eligible. Database searches were undertaken independently by two reviewers (CK and EP) with the results being compared. Once agreement was reached, duplicate records were removed, and titles and abstracts were screened independently by the same two reviewers. This was followed by full-text eligibility screening (CK and EP); where full-text publications were unavailable, corresponding authors were contacted, and a two-week response was permitted (all requests for full-texts were satisfied). During screening, any disagreements on eligibility were resolved by discussion; arbitration from a third reviewer was not required as consensus was reached on all disputed studies. Two reviewers (CK and EP) also hand searched through the full-text and reference lists of papers relevant to the topic (including editorials and reviews) for additional studies which met the eligibility criteria. When necessary, included studies which had multiple publications were linked together with the earliest paper of the trial used as the primary reference. Outcome data for these trials were extracted from the most comprehensive available data from the linked papers. 

Once the eligible papers had been determined, we were guided by Arksey and O’Malley’s [41] framework for scoping studies to help us determine the appropriate methods for charting the data. Accordingly, we extracted data relating to author(s), year of publication, geographical location, study population characteristics, details of the intervention and any comparators, other methodological considerations (Table 3), aims of the study, outcome measures, and key findings (Table 4). Once extracted, data were described in table format and qualitatively synthesised in the Section 3.

## 3. Results

### 3.1. Search Results

The database searches identified a total of 42 studies and a further three were identified from the full-text and/or reference lists of relevant publications. After removing duplicates (*n* = 9), 36 publications underwent title and abstract screening based on which a further 21 were excluded. A total of 15 studies were retrieved for full-text eligibility, and three were excluded with reasons (Figure 1). This left 12 papers which met the eligibility criteria; seven of these publications appear to be related to a single non-randomised trial [42,43,44,45,46,47,48]. Indeed, four of those seven cite the same institutional ethical approval reference [43,44,46,47], whereas, despite small differences in sample characteristics, the other three utilise the same methodology and it is assumed they are from one trial; whilst the authors did provide one full-text document, clarification requests to the authors were unsuccessful. The remaining five eligible papers all relate to individual RCTs [24,25,26,49,50].

### 3.2. Characteristics of Included Studies

Three of the included studies were conducted in Asia (Iran [25,50] and Pakistan [49]), whilst there was one study each conducted in Australia [26], Norway [24], and Brazil [48]. Two of the included trials were published in 2015 [24,48], two were published in 2016 [25,26], and one each from 2019 [50] and 2022 [49]. Despite variation in geographical location, all included trials used the Rotterdam PCOS criteria [3] to diagnose their participants and determine eligibility. The mean age of included participants ranged from 27 ± 5 [26] to 31.12 ± 2.42 years [50], whilst BMI ranged from 25.3 ± 1.96 [49] to 37.8 ± 11.4 kg/m^2^ [26], meaning that participants in all eligible studies were either overweight or obese. Participant characteristics are reported in Table 3.

All included studies incorporated a pre to post design to assess the effectiveness of a resistance training intervention in women with PCOS. Five studies [24,25,26,49,50] recruited only women with PCOS who were then randomised into an intervention or comparator group. In the remaining study [48], women with PCOS were compared to healthy controls and all participants received the intervention. The length of the applied intervention in the included studies ranged from 8 to 16 weeks (two trials [25,50] used eight weeks, two used 12 weeks [26,49] and one each used ten [24] or 16 weeks [48]). Regarding the applied training frequency, Vizza and colleagues [26] asked participants to exercise four times per week (two supervised and two home-based), whilst all other studies had a modal training frequency of three times per week (all supervised). 

Where training loads were reported, one study [24] used a fixed intensity for the duration of the intervention, whereas the remaining studies either progressed the intensity [25,48,50] or provided a target range [49]. One study did not specify the applied training load [26], but stated that the load was progressed along with gains in strength (Table 3). With regard to the number of sets and repetitions prescribed, three studies utilised three sets of either 10 [24], 8–15 [48], or 10–12 [49] repetitions. One study prescribed 1–2 sets of 15–20 repetitions [25], and another, 2–3 sets of 8–12 repetitions [26]; Hosseini and colleagues [50] did not specify sets/reps for land-based training, but utilised three sets of 12 repetitions for the water-based resistance training. Rest time between sets was only explicitly mentioned in three studies and were either one minute [24], two minutes [49], or an unspecified time, whilst a partner performed the exercise [48]. Similarly, session duration was only mentioned in four studies; total session time was stated as 60 min in two studies [26,48] or time spent performing resistance exercise in each session as 30 min [50] or approximately 32 min [49] in the remaining studies.

Whilst all interventions were broadly defined as either strength or resistance training, five studies [24,25,26,48,49] incorporated a range of dynamic exercises to work the major muscle groups and utilised a range of free weight (i.e., bench press, lunges, or deadlifts), body weight (i.e., push-ups or abdominal crunches), and machine-based (i.e., leg curl, leg extension, or chest press) exercises (Table 3). One study did not specify the exercises in their land-based intervention [50], but did state that a dumbbell was used for trunk strength training in their water-based arm. One study also prescribed calisthenics as a home-based intervention on non-resistance training days, which included three sets of 10 repetitions of body weight exercise (i.e., side leg raises, core stabilisation exercises, or wall squats) to facilitate habitual movement and behaviour change [26].

For those studies which incorporated comparator arms, two studies [26,49] had one additional comparison arm, two studies [24,25] had two additional arms, and the final study [50] (which had two eligible arms for this review) made four additional comparisons. Four studies [24,25,26,50] included an arm as a control that received no active intervention, two studies [24,49] included a high-intensity interval training (HIIT) arm, and two studies combined their resistance training interventions with pharmacological supplements; that is calcium [25] and vitamin D [50].

### 3.3. Outcome Measures

Across the included six studies, a range of outcomes relating to the health and wellbeing of women with PCOS are reported, including anthropometric variables [i.e., body weight, body mass index (BMI), body composition, etc.]. Four of these studies reported also changes to metabolic outcomes (i.e., fasting glucose/insulin, lipid profile) [24,25,26,48], four reported on changes to the androgenic profile of participants (i.e., testosterone, oestradiol, free androgen index, etc.) [24,26,48,49], and four reported on hormones associated with menstrual regularity (i.e., follicle stimulating hormone, luteinising hormone, anti-Mullerian hormone, etc.) [24,25,48,50]. Moreover, three of the included studies reported on outcomes relating to strength/hypertrophy (i.e., upper/lower body strength, arm/thigh muscle area, etc.) [25,26,48]. Only two studies reported on changes to cardiovascular health (i.e., resting heart rate, blood pressure, flow-mediated dilation, etc.) [24,48] and quality of life (QoL) [26,48], whilst one study each reported on sexual function [48] and physical activity levels [49]. It should be noted that the outcomes reported by Lara and colleagues, relate to seven papers [42,43,44,45,46,47,48], each with a specific focus (Table 4).

### 3.4. Effects of Resistance Training Interventions

When outcomes relating to the anthropometric measures of women with PCOS were assessed, two studies [26,48] reported statistical reductions in waist/umbilical waist circumference and/or waist-to-height-ratio, three studies reported statistical reductions in either fat mass or body fat percentage [24,48,49], whilst two studies reported an increase in fat free mass [24,48], one a decrease in the sum of skinfolds [48], and two showed a decrease in BMI [49,50]. Regarding reported metabolic outcomes, fasting blood glucose was statistically reduced in two studies [25,48], yet increased in another [26], whilst fasting insulin and the Homeostatic Model Assessment for Insulin Resistance (HOMA-IR) index were improved in one study [25]. The same study [25] was also the only one to report statistical improvements in triglycerides, total cholesterol and LDL-C; one further study [26] found statistical improvements in HbA1c. For studies which reported the effect of resistance exercise interventions on sex hormones, two studies reported statistical changes from baseline for serum testosterone [48,49], and another two for both circulating sex hormone-binding globulin (SHBG) levels and the free androgen index [24,48]. In addition, one study [48] reported statistical increases in androstenedione, and consequently adjusted the testosterone/androstenedione ratio; the same study reported statistical increases in prolactin too. Finally, two studies [24,50] found statistical reductions in the circulating levels of the anti-Mullerian hormone following resistance training interventions, with Hosseini and colleagues [50] reporting reductions for both land-based and water-based interventions.

For the studies reporting on outcomes relating to strength, three studies [25,26,48] reported improvements to lower body strength, two studies [25,48] for upper body strength, and one study for arm curl strength [48]; Lara and colleagues [48] also reported increased muscle area in women with PCOS. For outcomes referring to QoL, two studies [26,48] reported improved physical function, as measured by the 36-item Short Form Survey (SF-36); one of those studies [26] also reported improvements to the emotions and fertility domain scores from the health-related quality of life questionnaire for women with PCOS (PCOSQ), and lower levels of depression (as measured by the 21-items Depression, Anxiety and Stress Scale (DASS-21). Finally, one study [48] found improved sexual function, as measured by the Female Sexual Function Index (FSFI), and despite not reporting statistical significance, one study [49] reported a marked increase in PA, as measured by the International Physical Activity Questionnaire (IPAQ). No included study reported any measurable benefit to cardiovascular health.

## 4. Discussion

The aims of this scoping review were to identify and present all available evidence on resistance training interventions in women with PCOS and summarise key findings for any pertinent health-related outcomes. Overall, the findings of this scoping review suggest that resistance training is an effective treatment strategy for women with PCOS. Indeed, statistical improvements were identified for a range of pertinent outcomes in the management of PCOS, with many of these findings being replicated in multiple included studies. However, we also confirmed the paucity of relevant studies that have evaluated resistance training in women with PCOS, and notable limitations in the included studies (i.e., small sample size with negligible changes, as well as limitations regarding the study design and reporting).

### 4.1. Included Primary Studies

Given the high global prevalence of PCOS, the persistent lack of published studies returned from our database searching was surprising. Furthermore, when screening was completed, we anticipated that a larger proportion would be retained by our eligibility criteria. Whilst there are a very limited number of studies, most (*n* = 5) have adopted randomisation to allocate participants to intervention and control groups. That said, all individual studies were conducted from a single centre, and the RCTs have small sample sizes (resistance intervention arms ranged from 7–25 participants) meaning they may be insufficiently powered to detect between group differences [51]. The greatest number of statistical findings presented in this scoping review come from a case–control study [48] where the data has repeatedly been analysed to address multiple hypotheses. Whilst Lara and colleagues have the largest sample of women with PCOS completing an intervention, there are inherent limitations with this study design (e.g., risk of selection bias) that reduce our confidence in the certainty of the findings [52].

### 4.2. RT Training Composition and Reporting

In general, most resistance training components are reported to a reasonable standard. According to the FITT exercise prescription framework [53], all included studies report weekly training frequency and some measure of time. However, what is lacking in some studies is information on intensity and descriptions of the type of exercise being performed (Table 3). Explicit reporting of the components of complex interventions is essential for informing the reader about the content of effective programmes. Indeed, where reporting is insufficient, interpretation, replication, and implementation into clinical practice is impossible [54]. 

What is also lacking in these identified studies, are reports of fidelity to the intervention; only two trials [24,26] report the number of sessions completed by participants which makes it difficult to define the optimal dose [55]. Furthermore, description and implementation of control/comparator groups also detract from the certainty of this body of evidence. Given the nature of an exercise intervention, it is impossible to blind participants to their allocated group which may cause participant compliance and attrition issues, as they do not wish to be in the control group [56]. Furthermore, individuals who volunteer to participate in exercise-based studies are usually highly motivated to exercise, meaning that despite requests not to change their lifestyle habits, the control group may increase their PA levels anyway [57]. In the present scoping review, comparator arms are described with varying degrees of detail, but there is no mention of control group behaviours; this potential contamination may decrease the scope and statistical power to detect any true intervention effects [58].

### 4.3. Outcomes and Favourable Effects of Interventions

The included studies analyse a diverse range of outcomes that are pertinent to the management of women with PCOS. Encouragingly, many studies also report statistically favourable findings for multiple outcomes, and in many instances (i.e., blood glucose, testosterone, SHBG, strength, and fat free mass), included studies concur in their conclusions. It should however be noted, that there are some disparities between individual study findings (Table 4), and that in the vast majority of studies reporting statistical significance, the magnitude of changes from baseline are negligible and may not be clinically important [59]. Given the relatively short length of the resistance training interventions (8–16 weeks) in the included studies, it may be overly critical to expect larger, and clinically important changes in these outcomes, but equally none of the included studies completed follow-up assessments beyond the immediate end of the intervention to assess the persistence of reported benefits from the resistance training interventions [60] and whether the exercise behaviours had been sustained.

### 4.4. Strengths and Limitations of the Present Scoping Review

The present scoping review provides a broad overview of published studies that have assessed a resistance training-based intervention in the management of women with PCOS. Surprisingly, our systematic searches revealed a limited number of eligible studies, which involve a relatively small number of participants. By using broad terms and not specifying outcomes or study design, the search string we applied across databases was designed to capture a wide range of studies [61]. Whilst we are confident that we have identified all eligible studies in the extant literature, it is notable that three of the included papers were either returned from supplementary searches of relevant literature [25,50] or included studies [42]. This may be due to studies not being indexed in the databases we searched, or that they were written in another language; these factors should be considered in future searches of this body of literature.

Whilst we have adopted a critical approach in our appraisal of the included evidence, it may be useful for a full systematic review to be conducted which utilises pre-existing structured frameworks to evaluate constructs such as risk of bias [62,63], completeness of intervention reporting [64], and certainty of the evidence [65]. These were never the intended objectives for this scoping review; however, it may be beneficial for systematic reviews including these assessments to be completed, and also to include meta-analyses to assess the magnitude of pooled effects for each outcome.

### 4.5. Summary of Gaps in Literature

In the current body of literature regarding the use of resistance training interventions for the management of women with PCOS, we have identified a number of gaps which should be addressed in the future to better inform exercise recommendations for this population. (1) A limited number of primary studies exist, and those which are available have small sample sizes, meaning they are likely to be underpowered; a research priority for this population should be the implementation of a well-designed, clearly reported study that is sufficiently powered to detect meaningful changes. (2) A systematic review and meta-analysis which explicitly reports on the effectiveness of resistance training (either as a soul intervention, or in combination with other treatments) compared to alternative treatments is lacking from the evidence; such a systematic review should include quality assessment of the evidence according to the systematic review requirements/protocol. (3) None of the included studies completed follow-up assessments beyond the end of the intervention; adding follow-up measures to assess whether exercise behaviours and any associated benefits have persisted beyond the intervention should be a component of any future studies.

## 5. Conclusions

Resistance training programmes may be beneficial to the health and wellbeing of women with PCOS and may represent a viable method of exercise to those who are deconditioned or unable to tolerate aerobic exercise. However, the current published evidence to support this is limited, with few primary studies which typically incorporate small sample sizes (particularly RCTs), heterogenous sample characteristics, and varying degrees of exercise prescription. Whilst further steps (i.e., systematic review and meta-analysis) should be taken to further evaluate this body of evidence, based on the potentially positive evidence identified from this scoping review, it is apparent that there is a need for rigorously designed, multi-centred, and sufficiently powered RCTs so that certainty of effectiveness can be determined, and that findings can be generalised to the wider PCOS population.

## Figures and Tables

**Figure 1 medsci-10-00053-f001:**
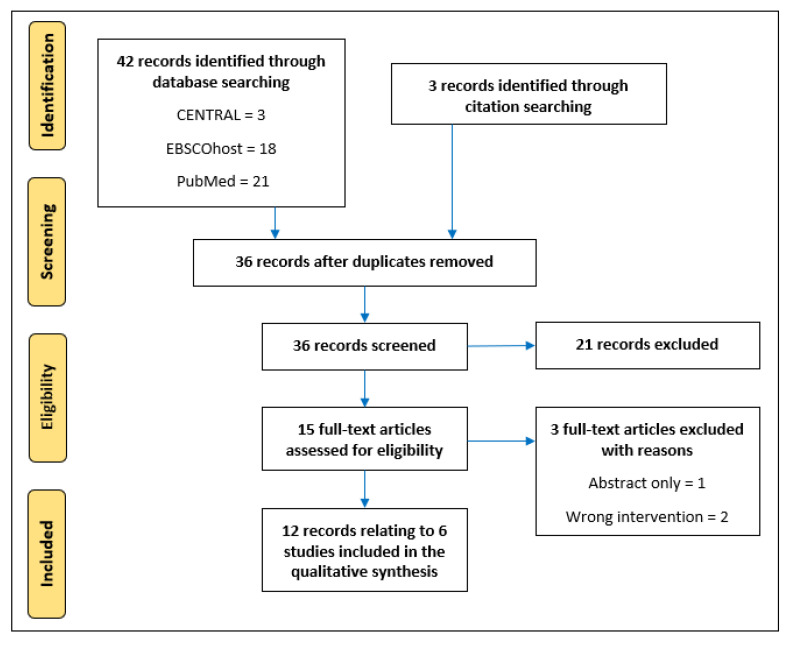
Flow diagram of database searches and study screening.

**Table 1 medsci-10-00053-t001:** Eligibility criteria for including studies in this scoping review.

Inclusion Criteria
1. Study design: studies incorporating a resistance training intervention; randomised controlled trials, cross-sectional studies, case–control studies are eligible 2. Participants: reproductive-aged women with a reported diagnosis of PCOS 3. Intervention: any study which incorporates an arm that utilises only resistance/strength training. Can be of any duration, supervised or unsupervised, with follow-up data collection of any duration 4. Outcomes: all reported outcomes
**Exclusion Criteria**
1. Study design: literature reviews, systematics reviews and meta-analyses, editorials, and commentaries 2. Participants: males, adolescent females, post-menopausal women Intervention: aerobic exercise, combined interventions (i.e., resistance training + aerobic training/diet/ pharmacological/etc. where effects of exercise cannot be isolated)

**Table 2 medsci-10-00053-t002:** Search algorithm for PubMed which was adapted for additional databases.

Search Algorithm for PubMed
(“Polycystic ovary syndrome” [MeSH Terms] OR “Polycystic ovar * [Title/Abstract]” OR “PCOS” [Title/Abstract] OR “PCOD” [Title/Abstract] OR “Stein levent *” [Title/Abstract] AND “Resistance Training” [MeSH Terms] OR “Muscle training” [Title/Abstract] OR “Strength training” [Title/Abstract] OR “Strengthening” [Title/Abstract])

* Commonly used wildcard symbol in PubMed which broadens a search by finding words that start with the same letters.

**Table 3 medsci-10-00053-t003:** Characteristics of studies included in this scoping review.

Study (year)	Study Characteristics	Participant Characteristics	Intervention	Comparator (s)
Vizza [26] (2016)	Design: RCT Location: Australia Sample size: 13 (resistance training: 7, control: 6) Diagnosis: Rotterdam PCOS diagnostic criteria	Age: 27 ± 5 years BMI: 37.8 ± 11.4 kg/m^2^	Duration: 12 weeks Frequency: 4 times/week (2 × RT, 2 home-based) Intensity: load not defined but progressed with strength gains. Two to three sets of 8–12 reps. Time: ~60 min per session Type: lat pulldown, leg curl, seated row, leg press, calf raise, chest press, split squat, shoulder press, biceps curl, triceps extension, and abdominal curl. Home-based: Callisthenics, 3 sets of 10 reps Participants were supervised for the resistance training but not for the home-based callisthenics	Control Participants did not receive any exercise intervention and were advised to continue with their current lifestyle, and usual healthcare and medical treatments.
Almenning [24] (2015)	Design: RCT Location: Norway Sample-size: 25 (resistance training: 8, HIIT: 8, control: 9) Diagnosis: Rotterdam PCOS diagnostic criteria	Age: 27.2 ± 5.5 years BMI: 26.7 ± 6.0 kg/m^2^	Duration: 10 weeks Frequency: 3 times/week Intensity: 75% 1-RM, 3 sets of 10 repetitions separated by 1 min rest Time: not specified Type: eight dynamic strength drills Participants were supervised	HIIT Frequency: 2 times/wk Intensity: 4 × 4 min at 90–95% HR_max_ separated by 3 min at ~70% HR_max_ Frequency: 1 time/wk Intensity: 10 × 1 min with maximal intensity separated by 1 min of rest/very low activity Type: treadmill, outdoor running/walking and/or cycling. Control Advised to adhere to the recommended ≥150 min per week of moderate intensity physical activity.
Lara [42,43,44,45,46,47,48] (2015 [48]; 2016 [42,44,46,47]; 2018 [45]; 2019 [43])	Design: Case-control Location: Brazil Sample size: PCOS: 45, Control: 52 Diagnosis: Rotterdam PCOS diagnostic criteria	PCOS Age: 28.1 ± 5.4 years BMI: 28.5 ± 6.02 kg/m^2^Control Age: 29.6 ± 5.3 years BMI: 26.2 ± 6.8 kg/m^2^	Duration: 16 weeks Frequency: 3 times/week Intensity: Progression from 60% 1-RM (week 1) to 85% 1-RM. Progression performed over 4 weeks of a microcycle; intensity increased, and volume reduced. Minimum of 3 sets of 8 repetitions Time: 60 min Type: bench press, leg extension, front latissimus pull-down, leg curl, lateral raise, leg press (45°), triceps pulley, calf leg press, arm curl, and abdominal exercise, executed in alternating segments. Participants were supervised.	All participants received the progressive resistance training intervention. Women with PCOS were compared to women without.
Rao [49] (2022)	Design: RCT Location: Pakistan Sample-size: 50 (resistance training: 25, HIIT: 25) Diagnosis: Rotterdam PCOS diagnostic criteria	Resistance trainingAge: 30.5 ± 4.8 years BMI: 25.3 ± 1.96 kg/m^2^HIITAge: 28.1 ± 4.9 years BMI: 26.5 ± 3.09 kg/m^2^	Duration: 12 weeks Frequency: 3 times/week Intensity: 60–70% 1-RM, 3 sets of 10–12 repetitions separated by 2 min of rest Time: ~32 min of work Type: squats, deadlifts, lunge, standing bent rowing, shoulder press, bench press, push-ups, and abdominal crunches Participants were supervised	HIIT Frequency: 3 times/wk Intensity: 4 × 4 min at 90–95% HR_max_ separated by 3 min of moderate intensity activity at ~70% HR_max_ Time: 45 min including warm-up and cooldown Type: treadmill
Hosseini [50] (2019)	Design: RCT Location: Iran Sample-size: 60 (control: 10, water training: 10, land training: 10, Vitamin D: 10, Water/Vitamin D: 10, Land/Vitamin D: 10) Diagnosis: Rotterdam PCOS diagnostic criteria	Water training Age: 31.12 ± 2.42 years BMI: 27.11 ± 0.74 kg/m^2^ Land training Age: 30.01 ± 1.70 years BMI: 27.01 ± 1.15 kg/m^2^ Control Age: 29.23 ± 2.11 years BMI: 26.70 ± 0.99 kg/m^2^ Vitamin D Age: 28.56 ± 1.55 years BMI: 26.71 ± 0.91 kg/m^2^ Water/Vitamin D Age: 29.43 ± 2.73 years BMI: 27.72 ± 1.47 kg/m^2^ Land/Vitamin D Age: 30.21 ± 1.65 years BMI: 27.86 ± 1.54 kg/m^2^	Duration: 8 weeks Frequency: 3 times/week Land training Intensity: 40% 1-RM progressing to 70% 1-RM at week 8 Time: 15 min warm-up, 30 min of resistance training, 5 min cooldown Type: weight training Water training Intensity: load not specified, 3 sets of 12 repetitions Time: 5–15 min warm-up, 60 min of resistance training, 15 min cooldown Type: trunk strength training with dumbbells Participants were supervised for both land- and water-based training	Control No specific detail provided Vitamin D Consumed Vitamin D3 supplement for 8 weeks, which was dosed by a physician according to the nature and severity of Vitamin D deficiency of subjects. Water/land training and Vitamin D These intervention arms followed the resistance training intervention whilst be supplemented with Vitamin D as above.
Saremi [25] (2016)	Design: RCT Location: Iran Sample size: 30 (resistance training and placebo: 10, resistance training and calcium: 10, control: 10) Diagnosis: Rotterdam PCOS diagnostic criteria	Age: 27.1 ± 5.1 years BMI: 25.5 ± 2.7 kg/m^2^	Duration: 8 weeks Frequency: 3 times/week Intensity: 40–60% 1RM of 1–2 sets of 15–20 repetitions Type: combination of free weights and machine weights including leg press, bench press, arm curl, and pulldown. Participants also took a placebo alongside the intervention (blinded). Participants were supervised	Control No specific detail provided. Resistance training and calcium Participants received 1000 mg per day of calcium alongside the resistance intervention.

Key: RCT: randomised controlled trial; BMI: body mass index; RT: resistance training; reps: repetitions; HIIT: high-intensity interval training; 1-RM: 1-repetition max; HR_max_: heart rate maximum; PCOS: polycystic ovary syndrome.

**Table 4 medsci-10-00053-t004:** Objectives, outcomes, and key findings from included studies.

Study (Design)	Study Aim(s)	Outcome Measures	Key Findings
Vizza [26]	To evaluate the feasibility of executing a randomised controlled trial of progressive resistance training in women with PCOS.	Body weight, BMI, waist circumference, fat mass, lean mass, fat-free mass, body fat (%), HbA1c, fasting insulin, fasting glucose, HOMA-2, hsCRP, testosterone, sex-hormone binding globulin, free androgen index, upper and lower body strength, PCOSQ (five domains: emotions, body hair, weight, infertility problems, and menstrual problems), SF-36 (eight domains: physical functioning, role physical, bodily pain, general health, vitality, social functioning, role emotional, and mental health), DASS-21 (three domains: depression, anxiety, and stress), and exercise self-efficacy scale.	For those performing progressive resistance training, there were statistical improvements in waist circumference, HbA1c, fasting glucose, lower body strength, and domains from the PCOSQ (emotions, infertility problems), SF-36 (physical functioning) and the DASS-21 (depression). By contrast, there were no statistical changes for any outcome in the control group. The authors concluded that a randomised clinical trial of progressive resistance training in women with PCOS would be feasible to conduct, and that there may be a beneficial effect on a range of key outcomes in this cohort. However, a suitably powered randomised controlled trial is required to confirm these findings.
Almenning [24]	To assess the effects of 10 weeks of structured exercise training on metabolic, cardiovascular, and hormonal outcomes in women with PCOS; the primary outcome measure was HOMA-IR. The comparison of high-intensity interval training and strength training (HIIT) was exploratory.	Body weight, BMI, waist circumference, fat mass, visceral fat, fat-free mass, VO_2_ max, resting heart rate, heart rate recovery, flow-mediated dilation, fasting glucose, fasting insulin, HOMA-IR, testosterone, free androgen index, anti-Mullerian hormone, sex-hormone binding globulin, dehydroepiandrosterone sulphate, cholesterol, HDL-C, LDL-C, triglycerides, homocysteine, hsCRP, adiponectin, and leptin.	For those completing the strength training intervention, there were statistically favourable effects for fat mass (%), fat-free mass (kg), free androgen index, anti-Mullerian hormone, and sex-hormone binding globulin. By contrast, HIIT improved fat mass (kg and %), VO_2_ max, flow-mediated dilation, fasting insulin, HOMA-IR, dehydroepiandrosterone, HDL-C, and homocysteine; there were no statistical changes in the control group. Observed changes were seen using exercise as a sole treatment (i.e., no dietary or pharmacological intervention) and without any changes in weight. Further research is needed to advance conclusions, and to establish exercise guidelines for these women.
Lara [42,43,44,45,46,47,48]	The study aimed to assess sexual function and emotional status of women with PCOS after 16 weeks of progressive resistance training [48]. To evaluate the efficacy of progressive resistance training for improving lean muscle mass, metabolic factors, and steroid hormones in women with PCOS compared to those without PCOS [44]. To investigate resistance training induced changes in telomere content and metabolic disorder in women with PCOS and controls [46]. To assess the effect of a 16-week programme of resistance training on the quality of life of women with PCOS [42]. To investigate the effects of periodized strength training on cardiac autonomic parameters and any correlation with metabolic/endocrine outcomes in women with PCOS [47]. To evaluate the effects of eight and sixteen weeks of progressive resistance training on body composition, indicators of hypertrophy, and muscle strength in women with and without PCOS [45]. Investigate the impact of progressive resistance training on obesity indices in women with PCOS and to assess the relationship between telomere length and obesity indices [43].	Female Sexual Function Index (six domains: desire, excitement, lubrication, orgasm, satisfaction, pain) [48]. Body weight, BMI, waist circumference, luteinising hormone, follicle stimulating hormone, oestradiol, androstenedione, testosterone, sex-hormone binding globulin, free androgen index, fasting glucose, fasting insulin, and HOMA-IR [44]. Body weight, BMI, waist circumference, body fat (%), fat-free mass, follicle stimulating hormone, luteinising hormone, prolactin, androstenedione, testosterone, oestradiol, sex-hormone binding globulin, free androgen index, glycaemia, fasting insulin, HOMA-IR, homocysteine, telomere length [46]. Body weight, BMI, waist circumference, and SF-36 (eight domains: physical functioning, role physical, bodily pain, general health, vitality, social functioning, role emotional, and mental health) [42]. Heart rate, systolic, diastolic, and mean blood pressure, body weight, BMI, body fat (%), testosterone, androstenedione, testosterone/ androstenedione ratio, sex-hormone binding globulin, free androgen index, fasting glucose, fasting insulin, HOMA-IR, and spectral analysis (supine and tilt test) [47]. Testosterone, androstenedione, sex hormone binding globulin, free androgen index, fasting glucose, insulin, HOMA-IR, arm muscle area, thigh muscle area, sum of skinfolds, body fat (% and kg), lean body mass, chest press, leg extension, and arm curl [45]. Body weight, prolactin, thyroid stimulating hormone, 17-hydroxyprogesterone, luteinising hormone, follicle stimulating hormone, oestradiol, androstenedione, testosterone, sex hormone binding globulin, free androgen index, glycaemia, insulin, HOMA-IR, homocysteine, telomere length, body fat (%), trunk body fat (%), android body fat (%), fat mass/height^2^, BMI, waist circumference, umbilical waist, waist-to-hip-ratio, waist-to-height-ratio, conicity index [43].	The sexual function of women with PCOS was statistically improved in all domains apart from orgasm and satisfaction whereas in the control women, statistical improvements were only seen in the pain domain [48]. In women with PCOS, following resistance training, waist circumference, testosterone, sex-hormone binding globulin, and glycaemia were statistically improved, whilst androstenedione concentration was increased. There were no significant differences in anthropometric characteristics, but values were consistently lower in women without PCOS pre-, and post-intervention [44]. Following progressive resistance training, women with PCOS had statistical reductions to waist circumference, body fat (%), testosterone, sex-hormone binding globulin, and free androgen index. Conversely, there were statistical increases in androstenedione, prolactin, and fat-free mass (kg) [46]. Following resistance training, the physical functioning of women with PCOS was statistically improved. No other domain reached statistical significance. By contrast, the control women saw statistical effects for vitality, social functioning, and mental health domains [42]. Women with PCOS who completed the resistance training intervention statistically reduced serum testosterone and exhibited changes to the testosterone/androstenedione ratio; there were no statistical changes in the control group. There were no statistical changes from baseline in the spectral analysis [47]. Women with PCOS had a statistical reduction in fasting glucose, testosterone, sex hormone binding globulin, sum of skinfolds, and body fat (% and kg). By contrast, androstenedione, arm and thigh muscle area, lean body mass, and weight moved during chest press, leg extension and arm curl exercises all increased. Control women also had similar benefits, particularly those relating to body composition and strength. Progressive resistance training was shown to increase strength parameters in women with PCOS and this is likely to the intrinsic hyperandrogenism associated with PCOS [45]. Following the resistance training intervention, women with PCOS had statistical changes to their biochemical profile: androstenedione, testosterone, and glycaemia all differed; these were the same for the control women too. Women with PCOS also had reductions in waist circumference, umbilical waist, waist-to-height-ratio, and conicity index [43].
Rao [49]	To evaluate the efficacy of high-intensity interval training (HIIT) on serum testosterone levels, body fat percentage, and physical activity levels among women with PCOS. HIIT was compared to a strength training intervention.	BMI, testosterone, body fat (%), and minutes per week of physical activity (IPAQ)	For those completing strength training there were favourable changes observed for all four outcomes; pre-post statistical significance is not reported in the results. Similar effects are observed for those completing the HIIT intervention but again, statistical significance is not reported. The authors statistically contrast change from baseline HIIT and strength training and report greater benefit from HIIT for all outcomes apart from BMI (no difference).
Hosseini [50]	The study aimed to investigate the effect of resistance training, in water and on land, with and without Vitamin D supplementation, on anti-Mullerian hormone levels in women with PCOS.	Anti-Mullerian hormone and BMI	Those performing resistance training (either on land or water) demonstrated statistical improvements in anti-Mullerian hormone and BMI. When interventions were performed in combination with Vitamin D supplementation, the effects were increased; these changes were not evident in the control group.
Saremi [25]	The study aimed to investigate the effect of eight weeks of resistance training, with and without calcium supplementation, on levels of anti-Mullerian hormone and metabolic parameters in women with PCOS.	Body weight, BMI, bench press, leg press, cholesterol, triglycerides, LDL-C, HDL-C, HOMA-IR, fasting glucose, fasting insulin, anti-Mullerian hormone.	Those who completed the strength training (and placebo) intervention had statistical change from baseline improvements for fasting insulin, fasting blood glucose, triglycerides, cholesterol, LDL-C, HOMA-IR, and upper/lower body strength; there was a statistical increase in body weight too. All these changes were also observed in the strength training and calcium combined group. The combined group reported statistical changes to anti-Mullerian, which were not observed in the strength training alone or control group.

**Key:** PCOS: polycystic ovary syndrome; BMI: body mass index; HbA1c: glycated haemoglobin; HOMA: homeostatic model assessment; IR: insulin resistance; hsCRP: high-sensitivity C-reactive protein; PCOSQ: polycystic ovary syndrome questionnaire; SF-36: 36-item short form health survey, DASS-21: 21-item depression, anxiety and stress scale; VO_2_ max: maximal oxygen consumption; HDL-C: high-density lipoprotein cholesterol; LDL-C: low-density lipoprotein cholesterol; IPAQ: international physical activity questionnaire.

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
