# Peer review of "Time to Load Up–Resistance Training Can Improve the Health of Women with Polycystic Ovary Syndrome (PCOS): A Scoping Review"

_medsci, 2022, doi:10.3390/medsci10040053_

Round 1

Reviewer 1 Report

Dear Authors,

The issue of the relationship between resistance training and health in women with PCOS is very interesting, and there is still a lack of high-quality studies on this issue.

The literature review presented by the authors of this study addresses this important medical problem, although ultimately it does not show effective solutions. I do not see any major substantive errors, but I have some doubts. 

Comments and Suggestions for Authors:

1. In my opinion, basing a scientific article on the results of a dozen studies by other authors (non-randomized; relating to six studies, from what period of time?, very diverse groups, e.g. mean BMI even 37.8)  does not contribute enough to the development of science.  Perhaps you should consider broadening the scope of the work and keywords and / or the range of publication years. I am asking the authors to respond to this opinion.

2. In the Introduction section, the authors accurately indicated a gap in the literature indicating the necessity of this research.

3. Keywords should be different than the words in the title (eg. polycystic ovary syndrome; PCOS).

Thank you for the opportunity to review this article.

Author Response

We would like to sincerely thank the Reviewer for the time dedicated to reviewing our work, and for the provided comments and suggestions. Please find a point-by-point response below.

Reply to point 1: Thank you for this comment/suggestion. We agree with the Reviewer that high-quality published research studies on this specific issue/field are lacking. As we also acknowledged this lack of robust evidence for resistance training as part of the lifestyle management of women with PCOS, in the submitted paper we completed a scoping review that identified and summarized the available published evidence on this specific issue/question. Such a scoping review has not been completed previously and, in our experience, this has also been highlighted as a gap in the existing scientific literature by relevant stakeholders when assessing proposals for supporting interventions with resistance training for women with PCOS. As the purpose of a scoping review is to collate and present to the readers/stakeholders all available published evidence, it is not uncommon for such scoping review papers to systematically summarize a limited number of published studies. Thus, our present paper can add to the literature the missing scoping review on this specific issue and contributes to the available literature by providing to readers/stakeholders this systematic summary/overview of the existing relevant studies. For that reason, we opted to conduct a scoping review on this specific issue (instead of a broader narrative review), and therefore, we followed all the required steps of a scoping review protocol. In our protocol and systematic searches, we used broad terms/keywords (please see lines 343-345 in the discussion of the revised manuscript), and we applied no limit for the range of publication years (please see amended lines 128-129 in the methods section of the revised manuscript). Therefore, we are confident that our scoping review has identified all key eligible studies on this specific issue/research question. We acknowledge the suggestion for broadening the scope of the presented work, however this would be outside the protocol of the present scoping review. As there may be studies that have not been indexed in the databases we searched (PubMed, CENTRAL, CINAHL and SportDiscus), we have also mentioned this point in the limitations section (however, it is unlikely that a key study on this specific issue is not indexed in any of the searched databases). Overall, we believe that the present scoping review addresses a gap in the relevant scientific literature and presents to the readers and relevant stakeholders the summary/overview of the available published evidence in order to inform the design of large and sufficiently powered RCTs which are required to clarify the role of resistance training as part of the lifestyle management of women with PCOS. In this context, and according to the point raised by the Reviewer, we have also added in the revised text of the manuscript, a summary of year of publication (please see lines 168-169 in the results section of the revised manuscript), and data on the heterogenous nature (i.e. age and BMI) of the study participants (please see lines 171-173 in the results section of the revised manuscript). We have also added further critique of the identified available evidence by suggesting that future systematic reviews on this specific issue/question should conduct quality analysis of the evidence (please see lines 368-369 of the revised manuscript) and by explicitly stating that the current evidence base involves a heterogenous sample and varying degrees of exercise prescription (please see lines 378-379 in the conclusion of the revised manuscript).  

Reply to point 2: Thank you for this comment. As mentioned for point-1, we agree with this comment, and because the lack of a relevant scoping review on this specific issue/question has also been highlighted as a gap in the literature, we opted to complete a scoping review on the specific issue of resistance training for the management of women with PCOS, rather than a broader narrative review which would not have the restrictions of a scoping review protocol.

Reply to point 3: Thank you for this suggestion. We have revised the keywords as follows: strength training; lifestyle; metabolism; hormones; quality of life; women’s health.

As an additional correction, in the revised manuscript we have also deleted a duplicate reference for the Vizza et al. 2016 paper and, accordingly, we have renumbered the references in the revised manuscript.

Reviewer 2 Report

This scoping review by Kite et al. sought to ascertain the state of research on the impact of resistance training (vs. aerobic training) in PCOS patients. The review was carefully defined, meticulously executed, and rigorously interpreted. The authors conclude that there is a general paucity of investigations into the potential benefits of resistance training on PCOS patients. Though the handful of studies they identify on this topic report promising results, they are not yet consistent enough to yield firm conclusions. Overall, this is a comprehensive summary of an important, but neglected, question that should encourage further research. I have no specific comments for improvement.

Author Response

We would like to sincerely thank the Reviewer for the time dedicated to reviewing our work, and for the provided supportive comments.  

Round 2

Reviewer 1 Report

Dear Authors,

Thank you for the opportunity to review this paper.

Thank you for taking into account my comments and relevant significant additions.

Thank you for your extensive and insightful response to my doubt # 1. I understand your opinion and your point of view, and I accept these arguments. I hope that, as expected, the manuscript will receive great interest (and citations).

Thank you.